# The Association between Difficulties with Speech Fluency and Language Skills in a National Age Cohort of Children with Down Syndrome

**DOI:** 10.3390/brainsci11060704

**Published:** 2021-05-26

**Authors:** Kari-Anne B. Næss, Egil Nygaard, Hilde Hofslundsengen, J. Scott Yaruss

**Affiliations:** 1Department of Special Needs Education, University of Oslo, 0318 Oslo, Norway; 2Department of Psychology, University of Oslo, 0373 Oslo, Norway; egilny@psykologi.uio.no; 3Faculty of Teacher Education, Arts and Sports, Western Norway University of Applied Sciences, 6851 Sogndal, Norway; Hilde.Hofslundsengen@hvl.no; 4Communicative Sciences and Disorders, Michigan State University, East Lansing, MI 48824, USA; jsy@msu.edu

**Keywords:** down syndrome, fluency, disfluency, co-occurrence, comorbidity, language

## Abstract

The present study (a) addressed difficulties in speech fluency in children with Down syndrome and typically developing children at a similar non-verbal level and (b) examined the association between difficulties with speech fluency and language skills in children with Down syndrome. Data from a cross-sectional parent survey that included questions about children’s difficulties with speech fluency, as well as clinical tests from a national age cohort of 43 six-year-olds with Down syndrome and 57 young typically developing children, were collected. Fisher’s exact test, Student’s *t*-test, linear regression, and density ellipse scatter plots were used for analysis. There was a significantly higher occurrence of parent-reported difficulties with speech fluency in the children with Down syndrome. Higher language scores were significantly associated with a lower degree of difficulties; this association was strongest for vocabulary and phonological skills. Although difficulties with speech fluency were not reported for all children with Down syndrome, a substantially higher occurrence of such difficulties was reported compared to that for typically developing children. The significant association between difficulties with speech fluency and the level of language functioning suggests that speech fluency and language skills should be taken into consideration when planning treatment for children with Down syndrome.

## 1. Introduction

A child’s level of speech fluency can affect effective communication [1]. Disfluent speech is common in young children during periods when speech, language, and emotional functioning progress rapidly [2,3]. One group that is reported to exhibit difficulties with speech fluency across ages is individuals with Down syndrome [4,5,6]. Down syndrome is the most commonly known single biological cause of intellectual disability [7,8]; it affects more than 1 live birth per 1000 [9]. Considerable risk of communication and language disorder has been observed in previous research with this group of children [10,11]. Variables that may be associated with language disorder in this group of children include varying extents of hearing loss, including repeated “otitis media with effusion” [12,13,14]; oral and palate conditions [15,16], including differences in the craniofacial structures and shape of the palate and hypotonic oral musculature [17]; and reduced cognitive functioning [18], including impaired auditory short-term memory [19,20]. The language profiles of children with Down syndrome commonly show a relative gap in expressive versus receptive language skills, favouring the receptive domain (c.f. [21]). Consistent weaknesses, compared to typically developing children of similar non-verbal mental age level, are reported in the areas of expressive vocabulary, receptive and expressive grammar (syntax and morphology; [22,23,24]), and phonological awareness ([25,26]; see also a systematic review and meta-analysis by Næss et al. [11]). Speech production, including speech fluency, is also commonly affected [27,28,29]. Although there is an initial gap between expressive and receptive language domains and between vocabulary and other core language skills, all of these areas develop more slowly over time in children with Down syndrome than in younger typically developing peers with similar non-verbal mental age levels, and the gap between the groups increases over time [26,30,31].

Previous research shows that the level of language functioning (see, e.g., the review by Ntourou et al. [32]) and dissociations across language domains may relate to fluency difficulties in typically developing children [33,34]. The high co-occurrence of Down syndrome and disfluency, combined with the specific language profile in children with Down syndrome (which includes a low level of language functioning and a gap between expressive and receptive language domains), suggests that such a link may also exist for children with Down syndrome. This question has not been thoroughly investigated in previous research, however. Thus, in the present study, we aimed to investigate whether there is an association between difficulties with speech fluency and language functioning in children with Down syndrome.

### 1.1. Difficulties with Speech Fluency in Children with Down Syndrome

Disfluencies of different types may interrupt the smooth flow of speech [35]. Some of these disfluencies may reflect a communication disorder such as stuttering (“childhood onset fluency disorder” in the DSM–5; [1]). Examples of stuttering-like disfluency include repetitions of sounds or syllables, prolongations in sounds, or blocks [3,36]. Nonstuttering disfluencies, also called “other” disfluencies [27,37], are experienced by most speakers. These include interjections, repetitions of multisyllabic words or phrases, and revisions [3]. Research suggests that children with Down syndrome exhibit all types of disfluencies [6,27], and they show more frequent stuttering-like disfluencies than other types of disfluencies [27]. Research indicates a higher occurrence of stuttering within individuals with Down syndrome than in both typically developing individuals [38] and individuals with intellectual disability due to other causes [39]. Very few studies have directly compared difficulties with speech fluency between children with Down syndrome and typically developing children. Instead, studies have reported only the occurrence of fluency difficulties within a group of children with Down syndrome, or they have used results from other studies of typically developing children as reference values for comparison with their own values measured from children with Down syndrome.

Estimates of the prevalence and incidence of stuttering in the otherwise typically developing population vary between 5% [40] and 11% [41], while in children with Down syndrome, the prevalence of stuttering varies between 10% and 47% [5,42]. The large apparent variation in results across studies focusing on individuals with Down syndrome may be due, in part, to the differences in the consideration of the types of speech disfluencies (see review [27]), the criteria used for diagnosing stuttering (c.f. [43,44]), and the languages spoken (c.f. [45]). In addition, methodological issues, such as small samples of individuals with Down syndrome (e.g., *N* = 28 in [46]; *N* = 26 in [27]; *N* = 1 in [47]; *N* = 5 in [48]) or the wide age range of the participants, may have impacted the results. Notably, the practice of including both children and adults in the same study sample (e.g., age ranging from 3.8 years to 57.3 years; [6]; see also the review by Kent and Vorperian [42]) is problematic due to the phenotype of Down syndrome. For example, neuropathologies characteristic of Alzheimer’s disease may already be pervasive in adults with Down syndrome by their 30s [49]. This may introduce a bias associated with the occurrence of difficulties with fluency, as language and communication are often reliably affected in this disease [50,51]. In particular, semantic verbal fluency has been found to be strongly associated with Alzheimer’s disease in individuals with Down syndrome [52].

To our knowledge, very few previous studies have investigated the occurrence of difficulties with speech fluency in samples consisting *only* of children with Down syndrome. Eggers and van Eerdenbrugh [27] are, as far as we know, the only one (ages 3.03–12.06 years). Salihovic et al. [53], Schieve et al. [38], and Wilcox [48] also investigated speech fluency in children, but they all had a mixed sample with teenagers. Notably, mixing these age groups or even mixing preschool-age children and school-age children may introduce uncertainties into the data and make it difficult to discern the true occurrence of difficulties with speech fluency in the population of children with Down syndrome. For example, in typically developing children, a higher occurrence of children with difficulties with speech fluency is suggested in preschool-aged children than in school-aged children [41,54]. This means that the wide age range in previous studies and the common lack of a typically developing comparison group could have biased the occurrence estimates and evaluation of difficulties with speech fluency in children with Down syndrome.

### 1.2. The Purpose of the Present Study

Although the evidence regarding the occurrence of difficulties with speech fluency in children with Down syndrome has limitations, the existing research results are generally consistent across several studies: children with Down syndrome are more likely to exhibit disfluent speech than other children. Language disorders resulting from a lower level of language skills and dissociations between the receptive and expressive language domains are also more apparent in this group of children. Together, these patterns lead to the hypothesis that there is a potential association between language functioning and disfluency in children with Down syndrome. However, there are uncertainties about the role that language development may play in the speech fluency of individuals with Down syndrome. The potential relationship between language and disfluency has not been thoroughly investigated in a sample of children with Down syndrome. In the current study, therefore, we studied a national age cohort of children with Down syndrome (and a group of typically developing children with similar non-verbal mental age levels) to ask the following research questions:(1)What is the occurrence of difficulties with speech fluency in a national age cohort of children with Down syndrome compared to that of a cohort of typically developing children at the same non-verbal mental age level?(2)What is the association between difficulties with speech fluency and language skills in children with Down syndrome?(3)Is there more dissociation in expressive and receptive language scores among children with Down syndrome who have difficulties with speech fluency compared to children with Down syndrome who have no difficulties with fluency?

Based on the uncertainties about the categorization of speech disfluency in individuals with Down syndrome in previous research, specifically whether the presence of disfluencies might reflect a fluency disorder such as stuttering [27,42], we focused on difficulties with speech fluency in general rather than the presumed diagnosis of a particular type of fluency disorder. In this way, the present data contribute to the overall understanding of the potential relationship between difficulties with speech fluency and language development in children with Down syndrome. Such information will contribute new knowledge related to assessment and treatment practices for children with cooccurring Down syndrome and difficulties with speech fluency.

## 2. Materials and Methods

The data included in this paper are original and obtained from a larger research project on language, reading, and communication skills in a national age cohort of 43 children with Down syndrome [26]. The Regional Committees for Medical and Health Research Ethics Sør-øst approved the study (reference ID: 19732), including the information letter and consent form, in advance.

### 2.1. Participants

The invitation letter and consent form were sent out by habilitation services across Norway to the registered parents of each child with Down syndrome who was scheduled to start school. Parents of 43 children with Down syndrome accepted the invitation and returned the consent form to the principal investigator. In their acceptance, they also confirmed that their child did not have any known comorbid diagnoses of autism spectrum disorder and that Norwegian was the child’s first language. Of the 43 children with Down syndrome, two participants were excluded because of missing data regarding difficulties with speech fluency. Thus, the final sample consisted of 41 participants (21 boys and 20 girls) with a chronological age of *M* = 75.79 months (*SD* = 3.57 months) and a raw non-verbal mental ability score (block design) of *M* =12.32 (*SD* = 5.51). In addition, parents of 57 typically developing children with similar non-verbal mental abilities accepted the invitation to serve as controls. These children were recruited from eight kindergartens in a Norwegian municipality; they were required to have Norwegian as their first language and no history of special educational needs. Of the 57 typically developing children, 3 participants were excluded due to missing data on the dependent variable. The final sample of typically developing children consisted of 54 participants (26 boys and 28 girls; chronological age: *M* = 36.50 months, *SD* = 4.15 months; non-verbal mental ability raw score (block design): *M* = 12.57, *SD* = 4.48).

### 2.2. Data Collection

Two sources of data collection were used: a parental questionnaire administered online and clinical tests. For the parental questionnaire, an email was sent to one parent of each participating child. Two reminders were sent out if no answers were received within the deadline. The answers were automatically coded in SPSS from the digital questionnaire. For the clinical test data, children were assessed individually and in person in three sessions. All test answers were registered manually in the standardized test protocol, and expressive answers were audio recorded for later verification.

### 2.3. Measures

All measures included in this sub-study are presented below. For all tests, standardized procedures for implementation and scoring were followed. In the scoring of expressive tests, the children were not penalized for systematic articulation mistakes. Internal consistency as a function of the number of test items and the average intercorrelation among the items from the full sample of 43 children participating in the main project are reported in brackets in the individual test descriptions.

#### 2.3.1. Difficulties with Speech Fluency/Stuttering

The dependent variable was assessed via the parent questionnaire language and reading development in children with Down syndrome (for a full English version of the questionnaire, see [55]). The parent was asked to rate their child’s “degree of difficulties with speech fluency/stuttering,” with a four-category answer option: from no difficulties with disfluency/stuttering (1) to a high degree of difficulties with speech fluency/stuttering (4). Difficulty with speech fluency/stuttering (hereafter called difficulties with speech fluency) was mainly analysed as a continuous variable, though it was dichotomized to investigate the last research question with “none” interpreted as indicating no difficulties with speech fluency and little, moderate, and high interpreted as indicating a difficulty with speech fluency.

#### 2.3.2. Non-Verbal Mental Ability

We used the Block Design subtest of the Wechsler Preschool and Primary Scale of Intelligence (WPPSI-III; [56]) as a background measure. In this non-verbal test, the child was asked to copy a building block pattern (shown with blocks or as a picture). The maximum score was 40, and the internal consistency of the block design was high across all 20 items (Cronbach’s α = 0.81).

#### 2.3.3. Vocabulary

Two tests of vocabulary were used: the Norwegian versions of the British Picture Vocabulary Scale (BPVS-II; [57,58]) and Picture Naming (WIPPSI–III; [56]). The BPVS-II is a receptive vocabulary test in which the examiner says a target word and the child is asked to point out the picture corresponding to the word among four pictures. The target words name animals, emotions, and professions, with increasing difficulties. The maximum score was 144, and the internal consistency of the BPVS was high across all 144 items (Cronbach’s α = 0.93). In the expressive vocabulary test, Picture Naming, the child was shown a picture of, e.g., a ball, a pencil, and an ambulance, and asked to name the item. The maximum score was 38, and the internal consistency of Picture Naming was high across the 38 items (Cronbach’s α = 0.90).

#### 2.3.4. Grammar

Two tests were used to assess grammar: the Norwegian versions of the Test for Reception of Grammar (TROG-R; [59,60]) and Grammatic Closure (Illinois Test of Psycholinguistic Abilities (ITPA; [61,62]). In the receptive test TROG-R, the examiner says a sentence, and the child is asked to point out the picture that corresponds best among four pictures. The maximum score was 80, and the internal consistency of the TROG was high across all 80 items (Cronbach’s α = 0.85).

The Grammatic Closure test is an expressive subtest from the ITPA, where the child must answer grammatically correct nouns, verbs, adjectives, prepositions, and possessive pronouns. For example, if the child looked at a picture, and the examiner read a corresponding ‘model’ sentence: “Here is one bed. Here are two…?”, the child’s task would be to finish the new sentence based on the ‘model’ sentence. The maximum score was 33, and the internal consistency of the Grammatic Closure was high across all 33 items (Cronbach’s α = 0.72).

#### 2.3.5. Phonology

To assess phonological awareness, we used four receptive measures adapted from Carroll et al. [63]: initial syllable matching, final syllable matching, rhyme matching, and initial phoneme matching. In each of the tasks, a puppet was used to make the assessment more child friendly. For example, when assessing the initial syllable, the child was told that the puppet likes to collect words that start with the same syllable. The puppet showed a picture card to the child and asked the child to point at the picture that began with the same syllable among two more picture cards on the table. The task was presented in the same way for the other three phonological awareness measures, with a different puppet for each measure. The maximum scores of both syllable measures are 8 each, while the maximum scores of the remaining two measures are 16 each. Cronbach’s α for initial syllable = 0.57, final syllable = 0.77, rhyme = 0.85, and phoneme = 0.83.

In addition to phonological awareness, expressive phonology was measured using a Norwegian version of the Children’s Test of Non-Word Repetition [64,65]. In this test, the child first heard a non-word of between two and five syllables in length and was asked to repeat the word. The maximum score was 28, and the internal consistency of the Children’s Test of Non-Word Repetition across all 28 items was high (Cronbach’s α = 0.83).

#### 2.3.6. Speed of Processing

Speed of processing was measured by rapid automatized naming (RAN) and Child Language and Learning’s speed of processing tests [66]. Two tasks with black and white drawings of objects were used to assess expressive processing speed. The objects represented high-frequency words usually acquired at a very early age, such as RAN1: a sun, a boat, a mouse, a door, and a bus; and RAN2: a light, a ball, a boy, a house, and a car. The five pictures were shown randomly in four rows with five items in each row. The child was asked to name each picture. The time it took to complete the task was recorded, and mean summary scores were calculated for the total amount of time used on the two tasks. The intraclass correlation between RAN1 and RAN2 was moderate (ICC = 0.57; when using a two-way mixed effects model with absolute agreement based on an average of the two measures [67].

Two tasks were used to assess the receptive speed of processing: both involved focusing on objects; the words used were high frequency and usually acquired at an early age [66]. In the first speed of processing task (SPEED 1), the child was given a sheet of paper showing black and white drawings of a sun, a boat, a mouse, a door, and a bus. The five pictures were shown randomly in six rows with seven items in each row. There were four sheets in total. For each sheet, the child was shown a mouse and asked to collect all the mice on the sheet. Then, the child was given a marker and asked to set a dot on all the mice on the sheet. Finally, the child was asked to do the task as quickly as possible for one minute. The number of tasks that the child completed correctly within the time frame was summarized. In the second speed of processing task (SPEED 2), the child carried out the same task as in SPEED 1, but the pictures were of a light, a ball, a boy, a house, and a car, and the child was asked to collect cars. The scoring scheme for SPEED 2 was the same as that for SPEED 1. The mean summary scores were calculated for the total amount of time taken to complete the two speed tasks. The intraclass correlation between SPEED 1 and SPEED 2 was good (ICC = 0.87; when using a two-way mixed effects model with absolute agreement based on an average of the two measures).

### 2.4. Analysis

In total, six values out of 820 possible scores (0.7%) for the predictors used in the regression analyses were missing among the children with Down syndrome on the SPEED tasks because some children did not want to do those tasks. The results from Little’s test (chi-square (35) = 31.24, *p* = 0.65) indicated that these missing data were random, so the missing data were replaced by multiple imputation (50 datasets). All analyses and results, except for descriptive statistics (Table 1), were based on the data set that included these imputed data.

For the first research question, Fisher’s exact test, Student’s *t*-test, and linear regression analyses were used to test differences between children with Down syndrome and typically developing children. For the second research question, we combined receptive and expressive functioning within four domains: vocabulary, grammar, phonology, and processing speed. The associations between these four functional linguistic domains and the degree of difficulties with fluency were analysed with three levels of linear regression models: a bivariate model, a model controlling for non-verbal mental functioning and a full model including all four functional linguistic domains and non-verbal mental abilities as predictors. All variables were standardized (Z-values) before being combined, and all variables were again standardized before being entered into the regression models. Thus, the presented regression coefficients can be interpreted as standardized regression coefficients. For the third research question, we created disparity variables in which expressive functioning scores were subtracted from receptive functioning scores within each of the four functional domains (vocabulary, grammar, phonology, and processing speed). A total disparity variable was also calculated across all four domains. Again, all variables were standardized before being deducted, and the combined variables were standardized before being entered into linear regression analyses. The disparity variables were analysed as predictors for the degree of difficulties with fluency in bivariate analyses, controlled for non-verbal mental functioning, and in a full linear regression model with all functional linguistic domains and non-verbal mental functioning entered as independent variables. In addition, we investigated whether the confidence intervals for the regression coefficients for receptive and expressive functioning overlapped when they were entered separately into the model instead of the disparity variable. We used the Lmatrix function in general linear models to investigate whether the regression coefficients of receptive and expressive functioning were significantly different. We also analysed whether there is more dissociation in expressive and receptive language scores among children with Down syndrome who have difficulties with fluency compared to children with Down syndrome who have no difficulties with fluency in a similar manner to that done by Anderson et al. [34]. Specifically, we used density ellipse scatter plots to identify participants outside the 95% ellipse who also had a dissociation of more than 1 standard deviation between receptive and expressive scores.

The distribution of data was evaluated by analysing the residuals of the final regression models through histograms, scatterplots, and P-P plots. Multicollinearity was investigated through a correlation matrix and by the variance inflation factor (VIF). All analyses were performed using IBM SPSS Statistics version 27, with the exception of density ellipse plots, which were made with the package ggplot2 using R version 4.0.3. A significance level of 5% was chosen for all analyses. No a priori correction for multiple comparisons was made due to this being an exploratory observational study [68].

## 3. Results

### 3.1. Descriptive Statistics

The sex distribution was quite similar between samples, with 20 (49%) girls in the group with children with Down syndrome and 28 (52%) in the group of typically developing children (chi-square 0.088, *p*_exact_ = 0.84). There were no significant differences in children’s non-verbal mental functioning (*M* = 12.3, *SD* = 5.5 and *M* = 12.6, *SD* = 4.5 for children with Down syndrome and typically developing children, respectively, t(93) = 0.25, *p* = 0.80). Descriptive language data for the children with Down syndrome are presented in Table 1.

### 3.2. Research Question 1: Occurrence of Difficulties with Fluency in Children with Down Syndrome and Typically Developing Children

The distribution of the parent-reported difficulty with speech fluency is presented in Table 2. If dichotomizing the symptoms, 29 (71%) of the children with Down syndrome were judged to have difficulties with speech fluency, compared to 8 (15%) of the typically developing children. The difference in the difficulties with fluency between children with Down syndrome and typically developing children was highly significant, independent of whether levels of symptoms were dichotomized (chi-square = 30.65, *p* < 0.001) or were used as continuous variables, both before (β = 0.62, *p* < 0.001) and after (β = 0.61, *p* < 0.001) controlling for non-verbal mental functioning.

### 3.3. Research Question 2: The Association between Difficulties with Fluency and Language Skills in Children with Down Syndrome

To investigate the association between language skills and difficulties with speech fluency, we created four variables representing the four functional linguistic domains of vocabulary, grammar, phonology, and processing speed. Each variable reflected the mean of standardized (Z) values of the receptive and expressive tests for each domain. Table 3 presents the results from bivariate linear regression analyses for each of these skills when controlling for non-verbal mental functioning and a full model with both non-verbal mental functioning and all four functional linguistic domains as predictors. Vocabulary skills were significantly related to difficulties with speech fluency in all models, with moderate [69] effect sizes (*β* between 0.52 and 0.61). Grammar, phonology, and processing speed had small to moderate effect sizes in bivariate analyses (*β* between 0.30 and 0.40) and when controlled for non-verbal mental functioning (*β* between 0.26 and 0.38). However, of the three, only phonology skills were significant when controlling for non-verbal mental functioning. The effect sizes for grammar, phonology, and processing speed were negligible when all four domains were included in the model. There were no indications of any violations of assumptions for linear regression analyses for the full model, and the highest VIF was 3.1, indicating that there was not a high degree of collinearity. Nevertheless, correlations between the four functional domains (r from 0.36 to 0.72) may have influenced the results in the full model (see correlation matrix in Appendix A). As indicated above, no a priori correction for multiple comparisons was made. Nevertheless, the effects of vocabulary and phonology found in Table 3 are still significant after controlling for four comparisons [70].

### 3.4. Research Question 3: The Dissociation in Expressive and Receptive Language Scores between the Groups

To investigate the dissociation in expressive versus receptive functioning among children with Down syndrome, we created new variables presenting the discrepancy between receptive and expressive functioning. To ensure comparability across measures, all variables were standardized (Z-values) before expressive functioning was deducted from receptive abilities. No violation of assumptions for regression analyses or collinearity were found. Table 4 presents the association between these differences and the level of difficulties with fluency. Neither of the investigated domains of vocabulary, grammar, phonology, or processing speed nor the total receptive versus expressive difference were related to the level of difficulties with fluency.

In addition, we investigated whether the regression coefficients for receptive versus expressive functioning overlapped when both were entered into linear regression analyses. In this analysis, non-verbal mental functioning was controlled for, and the level of difficulties with fluency was the dependent variable. For all five comparisons, the confidence intervals for receptive and expressive functioning highly overlapped. Thus, none of the five contrasts were significant when comparing the receptive and expressive regression coefficients after controlling for non-verbal mental functioning using the transformation coefficients matrix (MMATRIX) function in general linear models (GLMs).

We further analysed the material in a similar manner as previously done by Anderson et al. [34]. Table 5 gives an overview of cases that met both requirements for dissociation. There was a general tendency for more dissociation among children with low levels of difficulties with fluency than among children with moderate or high levels of difficulties, but this was only significant for grammar (*p* = 0.05).

## 4. Discussion

The results showed a significantly higher occurrence of difficulties with speech fluency in children with Down syndrome than in typically developing children with similar non-verbal mental age levels (corresponding to a chronological age of ca. 3 years). In addition, a large percentage of children with Down syndrome were rated as having serious difficulties with speech fluency. This stands in contrast to the finding that none of the typically developing children showed a serious degree of difficulties with speech fluency. The associations between language measures and the degree of difficulties with fluency in the children with Down syndrome were significant for all language domains included in the bivariate analysis; higher language skills were associated with a lower degree of difficulties with fluency. After taking into account non-verbal mental abilities, vocabulary and phonological skills were still significantly associated with the degree of difficulties with speech fluency. However, the dissociation hypothesis, that is, that there is a relationship between more fluency difficulties and larger gap between expressive and receptive language domains, was not supported by the data.

### 4.1. High Occurrence of Children with Difficulties with Speech Fluency

The high occurrence of difficulties with speech fluency in children with Down syndrome compared to typically developing children at the same non-verbal mental age level was expected based on inferences drawn from existing research. No previous studies have exactly investigated the occurrence of difficulties with speech fluency in children with Down syndrome compared to typically developing children at the same non-verbal mental age level. However, our results pattern align with the results from a survey study that used a group comparison design to investigate the occurrence of fluency disorders. Schieve et al. [38] included a sample of 27 individuals with Down syndrome and 1393 typically developing individuals and found occurrences of 15.6% and 1.5%, respectively. The results also align with results from an audio sample study comparing separate estimates of the occurrence in individuals with Down syndrome to estimates in previous research on the occurrence in typically developing individuals [27]. They found an occurrence of 31% in children with Down syndrome, which stands in stark contrast to the commonly cited values of a 1% prevalence of stuttering in typically developing individuals [35] and a lifespan incidence of more than 5% [71].

In addition to aspects of speech and language skills, various other developmental aspects of physical abilities and psychological state have been suggested to explain unique variance in the development of stuttering in otherwise typically developing children [2,72]. Nevertheless, such multifactorial models were not developed to explain difficulties with speech fluency in general, and they were not designed for unique populations such as children with Down syndrome. It is apparent from prior research that children with Down syndrome have challenges with all of these aspects of development, including language [26,73], speech motor skills [16,30], and emotionality (e.g., [74,75]). The complex developmental profile found in children with Down syndrome, including a range of different challenges, may also make these children vulnerable to developing difficulties with speech fluency.

### 4.2. More Serious Difficulties with Speech Fluency

The degree of parent-reported difficulties with speech fluency in our sample of children with Down syndrome varied from no difficulty to severe difficulty. For typically developing children, parents reported only no difficulty or a small degree of difficulty. These results imply more variation in the difficulties across children with Down syndrome than in the (younger) typically developing children. These results are in line with the results from Eggers and Van Eerdenbrugh [27], who also showed a large variation in the percentage in both stuttering-like disfluencies and other disfluencies across their sample of children with Down syndrome. However, in the current study, the differences in chronological age between the two participant groups may have influenced the results. This is because the parents may have rated their child’s difficulties with fluency with peers at similar chronological age in mind. Due to the age effect of difficulties with fluency [71,76], the parents of the (older) children with Down syndrome may have an expectation of fewer difficulties with fluency in their children than the parents of the (younger) typically developing children whose age-matched peers may also have more disfluencies.

In the current study, some children with Down syndrome were rated to have *no* difficulties with speech fluency. In contrast, Eggers and Van Eerdenbrugh [27] reported that all of the children in their sample showed some disfluency. This difference in results may reflect that the current study focused on *difficulties* with speech fluency, while the study by Eggers and Van Eerdenbrugh [27] focused on the presence of a range of different types of speech disfluencies. This would allow a child to exhibit disfluencies without being judged by the parent to experience *difficulties* with fluency.

### 4.3. An Association between Difficulties with Speech Fluency and the Level of Language Skills

The results of the current study showed that better language skills are associated with a lower degree of difficulties with speech fluency. To the best of our knowledge, no other studies have investigated the association between language level and difficulties with speech fluency in *children* with Down syndrome. However, in typically developing children, associations have been reported between language skills and disfluency [77] and between language skills and stuttering [33,78,79,80]. Luckman et al. [79] found that children who stuttered scored almost one standard deviation below children who did not stutter on expressive vocabulary. In a range of studies, increasing the length and complexity of utterances has been found to be associated with increased stuttering in children [34,81,82,83,84,85,86,87,88,89,90,91,92,93]. Children who stutter are also shown to have increased difficulties with fluency on both monosyllabic function words [94] and unfamiliar words (non-words/novel phonological sequences) [95].

Children with Down syndrome usually have a broad language disorder affecting both sentence-level and word-level production. On average, they reach the milestone of sentence production at approximately 3.5 to 5 years of age [96,97], but 30% of children with Down syndrome still do not speak in complete sentences by the age of 6 [29]. For children who do speak in sentences, limitations in syntax and complex sentence structure are still reported [29,98], and they also have a low mean length of utterance [99]. In general, children with Down syndrome have limited expressive vocabulary [30] and show initial weaknesses in function words such as prepositions, conjunctions, and pronouns [100], as well as on unfamiliar words [24,26,101]. Children with Down syndrome may therefore be specifically vulnerable to difficulties with fluency due to aspects related to their expressive language skills, even though they may also be at a stage in their language development when they are still producing relatively simple sentences.

The association between difficulties with speech fluency and language in typically developing children who stutter has been the focus of a longstanding debate (e.g., [32,102,103]). The fact that language is a common active ingredient in existing treatment programs for stuttering [104] also suggests an association between difficulties with speech fluency.

### 4.4. More Difficulties with Speech Fluency Not Related to Higher Level of Dissociation in Expressive and Receptive Language Skills

The children with Down syndrome in this study had, on average, both a low language level and a dissociation between expressive and receptive domains. Nevertheless, the results do not indicate an association between a higher degree of difficulties with speech fluency and a larger gap between expressive and receptive language skills, with the only marginally significant finding actually going in the opposite direction. Contradictory to our findings, studies with typically developing children suggest that gaps in performance within or between linguistic subcomponents, such as between receptive and expressive vocabulary, are associated with stuttering [33,34,83,105]. Anderson et al. [34] concluded that their sample of 45 children who stutter (age 3–5.11 years) was three times more likely to have dissociations across speech-language domains than their sample of 45 children who do not stutter (age 3–5.11 years). Coulter et al. [83] replicated the paper from Anderson et al. [34], and their results showed that children who stutter were five times more likely to have dissociations than children who do not stutter. They suggested that the dissociations could be markers of speech and language production systems that are not congruent with each other [83]. However, we did not find a relationship between higher level of dissociation and more difficulties with speech fluency in children with Down syndrome. This may be due to a minimal impact of the gap between receptive and expressive skills on difficulties with speech fluency or to the coarse assessment of difficulties with fluency and the parent-reported nature of the variable.

### 4.5. Limitations

A number of limitations of this study should be mentioned. First, the sample size of the study reduced the number of associated variables that could be included in the analysis. Although this is a relatively large study on difficulties with fluency in children with Down syndrome, the number of participants was still low for statistical analysis. To reduce the possible bias from adding too many covariates into the analysis, we summed the scores of two or more variables, but this may have the unintended effect of diluting the relevant contributions of individual variables.

To keep the sample as large as possible and to investigate an unselected sample of children with Down syndrome, no selection criteria were imposed to facilitate convenience in the recruitment process in the current study. Eggers and Van Eerdenbrugh [27] had an original sample of 50 participants, but the number of participants reported in their paper was 26. Their selections may have been based on the number of utterances/syllables available for each child. Obtaining a sufficiently large speech sample may be difficult in this clinical group, particularly when seeking to obtain enough syllables of speech to decide whether children stutter or not. Notably, professional coding of the disfluencies of children with Down syndrome may be challenging due to large variations in the speech produced and their phenotypic characteristics, including pauses and varying speech rates [106,107,108,109]; difficulties with prosody, including differences in lexical stress, producing questioning intonation, and the use of imitating intonation [73]; and articulation difficulties [28,110]. The present results therefore complement the results from Eggers and Van Eerdenbrugh [27] by adding information about *parental* judgements, which consider context and experiences. Tumanova et al. [3] highlighted that parents’ report of difficulties with fluency in typically developing children is usually valid. On the other hand, it may be hard for parents of children with Down syndrome to evaluate their child’s difficulties with fluency independent of their child’s other complex speech, language, and communication disorders. Consequently, parents may have responded about language skills more generally than would an expert in speech and language therapy. The significant relationship between difficulties with fluency and language skills may therefore have been influenced by parents not clearly separating these two issues. Future research can supplement the knowledge base further by combining both parental and clinician judgements in evaluating the difficulties with fluency in children with Down syndrome.

The use of parental reports of difficulties with fluency status gives information about the difficulties with speech fluency across settings and partners but has limitations due to a lack of information about the severity and the types of disfluency—and whether the difficulties reflect an actual fluency disorder. In addition, it has been suggested that the difficulties with speech fluency in children with Down syndrome may represent a specific disfluency profile that does not fully overlap with the distribution of disfluencies in typically developing children who stutter [27]. This study adds knowledge related to difficulties with the fluency of a national age cohort and its association with language skills. It also confirms that parents’ judgements are able to identify variations in the degree of difficulties with fluency within a sample of children with Down syndrome. However, checking the data against clinical judgements will be of importance, and future studies should ensure the inclusion of a set of measures in addition to the parent’s judgements. Examples of data to be included in such inquiries include a clinical evaluation of typology, frequency, and severity of disfluencies across different speaking situations. There may also be some effects from fluency and/or language treatment that should be considered when the occurrence of difficulties with fluency are investigated in future research.

## 5. Conclusions

The results of parental data from this national age cohort of children with Down syndrome within a narrow age range indicate a significantly higher occurrence of difficulties with speech fluency compared to typically developing children of the same non-verbal mental age level. A significant association between difficulties with speech fluency and the level of language skills was discovered and should be taken into consideration when planning treatment for children with Down syndrome.

To date, limited research results on interventions and treatment of difficulties with fluency or fluency disorders for children with Down syndrome exist, and no effect study (e.g., a randomized controlled trial) is known to the authors. Until we know more about what constitutes effective treatment for this group of children, the large co-occurrence between difficulties with fluency and low language skills in children with Down syndrome supports a need for speech and language therapy that aims to simultaneously improve the child’s language development and speech fluency. Speech-language pathologists also have a responsibility for treating the complex communication disorder of this group of children. To tailor the treatment to research-based knowledge, future effect studies should be designed, especially for children with Down syndrome; these studies should control for language level to investigate the potential effects of fluency treatment on language development.

## Figures and Tables

**Table 1 brainsci-11-00704-t001:** Descriptive statistics.

	Children with No Disfluency (*n* = 12)			Children with Disfluency (*n* = 29)			Test of Difference (*t*-Test)
	Mean	*SD*	Range	Mean	*SD*	Range	*p*-Value
Non-verbal mental functioning	13.00	5.19	4–22	12.03	5.70	0–22	0.616
BPVS	28.33	13.06	2–50	20.69	10.33	1–37	0.053
Picture naming	10.17	5.44	1–16	7.48	5.65	0–20	0.170
TROG	9.25	6.61	1–27	8.62	5.07	0–19	0.743
Grammatic closure	2.08	2.54	0–6	1.10	1.74	0–6	0.161
Phonological awareness	24.17	8.54	5–41	17.24	10.79	0–31	0.055
Non-word repetition	3.17	2.48	0–7	2.55	3.50	0–10	0.584
Speed	9.17	5.32	0–17.50	7.55 ^1^	5.73	0–22.50	0.413
RAN	49.26	43.44	0–179.5	48.65	27.73	0–103.50	0.957

Note ^1^
*n* = 27. Non-verbal mental functioning was assessed with the Block Design subtest. Language functioning was assessed with British Picture Vocabulary (BPVS) and Picture Naming for vocabulary, the Test for Reception of Grammar (TROG) and Grammatic Closure subtest from ITPA for grammar, sum of four Phonological awareness tests and the Children’s Test of Non-Word Repetition for phonology skills, and the mean of two Child Language and Learning’s speed of processing tests (Speed) and the mean of two Rapid Automized Naming tasks (RAN) for processing speed.

**Table 2 brainsci-11-00704-t002:** Degree of difficulties with fluency among children with Down syndrome and typically developing children at the same non-verbal mental age level.

Degree of Difficulties	Children with Down Syndrome (*n* = 41) Mean (*SD*)		Typically Developing Children (*n* = 54) Mean (*SD*)	
	*n*	%	*n*	%
None	12	29	46	85
Small	10	24	8	15
Moderate	13	32		
Severe	6	15		

**Table 3 brainsci-11-00704-t003:** Regression analyses of association with difficulties with fluency among children with Down syndrome (*N* = 41).

	Bivariate Model				Controlled for Non-Verbal Mental Functioning				Full Model			
	*B*	95% CI of *B*	*SE*	*p*	*B*	95% CI of *B*	*SE*	*p*	*B*	95% CI of *B*	*SE*	*p*
Vocabulary	−0.52	−0.79, −0.26	0.14	<0.001	−0.60	−0.93, −0.28	0.17	<0.001	−0.61	−1.10, −0.11	0.25	0.016
Grammar	−0.32	−0.61, −0.02	0.15	0.038	−0.31	−0.69, 0.07	0.19	0.109	0.06	−0.38, 0.50	0.22	0.777
Phonology	−0.40	−0.68, −0.11	0.15	0.007	−0.38	−0.70, −0.06	0.16	0.020	−0.07	−0.51, 0.38	0.23	0.776
Processing speed	−0.30	−0.60, 0.00	0.15	0.053	−0.26	−0.59, 0.07	0.17	0.127	0.03	−0.35, 0.41	0.19	0.879

Note. The results from regression analyses were all standardized (Z-values) before being entered into the models. Thus, *B* can be interpreted as a standardized regression coefficient. The results for intercepts and non-verbal mental functioning are not shown, as these are not the subject of the present article. The full model includes non-verbal mental functioning and four variables combined from both receptive and expressive features: vocabulary, grammar, phonology abilities, and processing speed. Non-verbal mental functioning was assessed with the Block Design subtest. Receptive and expressive functioning were assessed with the British Picture Vocabulary and Picture Naming for vocabulary, the Test for Reception of Grammar and Grammatic Closure subtest from ITPA for grammar, the mean of four Phonological awareness tests and the Children’s Test of Non-Word Repetition for phonological skills, and the Child Language and Learning’s speed of processing tests and the Rapid Automized Naming task for processing speed. Data are based on 50 multiple imputed datasets for 0.7% missing data.

**Table 4 brainsci-11-00704-t004:** Association between difficulties with fluency and the gap between receptive and expressive skills in various language areas (*N* = 41).

Language Area	Bivariate Model				Controlled for Non-Verbal Mental Functioning				Full Model			
	*B*	95% CI of *B*	*SE*	*p*	*B*	95% CI of *B*	*SE*	*p*	*B*	95% CI of *B*	*SE*	*p*
Vocabulary	−0.03	−0.34, 0.28	0.18	0.850	−0.02	−0.33, 0.29	0.16	0.899	0.03	−0.33, 0.40	0.18	0.852
Grammar	0.07	−0.25, 0.38	0.16	0.678	0.10	−0.21, 0.41	0.16	0.534	0.19	−0.16, 0.55	0.18	0.280
Phonology	−0.21	−0.51, 0.10	0.16	0.191	−0.21	−0.51, 0.10	0.16	0.177	−0.30	−0.66, 0.06	0.19	0.104
Processing speed	−0.06	−0.37, 0.26	0.16	0.731	0.02	−0.32, 0.35	0.17	0.923	−0.05	−0.40, 0.29	0.18	0.762
Total	−0.04	−0.35, 0.28	0.16	0.820	0.00	−0.32, 0.32	0.16	0.993				

Note. Linear regression analyses of the association between difficulties with fluency and the differentiation between receptive and expressive functioning (receptive minus expressive). All variables were standardized before being combined, and all variables were again standardized before being entered into the regression models. The full model includes non-verbal mental functioning and four variables with the differentiation between receptive and expressive features: vocabulary, grammar, phonology, and processing speed. Non-verbal mental functioning was assessed with the Block Design subtest. Receptive and expressive functioning were assessed with British Picture Vocabulary and Picture Naming for vocabulary, the Test for Reception of Grammar and the Grammatic Closure subtest from the ITPA for grammar, the mean of four Phonological awareness tests and the Children’s Test of Non-Word Repetition for phonological skills, and the Child Language and Learning’s speed of processing tests and the Rapid Automized Naming task for processing speed. The total is the combination of the previous four domains. Data are based on 50 multiple imputed datasets for 0.7% missing data.

**Table 5 brainsci-11-00704-t005:** Cases with dissociative scores outside the density ellipse (*N* = 41).

	Children with No Difficulties with Fluency (*n* = 12)		Children with Difficulties with Fluency (*n* = 29)		Test of Difference Chi-Square
	*n*	%	*n*	%	*p*-Value
Vocabulary	0	0%	3	10%	0.543
Grammar	4	33%	2	7%	0.050
Phonology	1	8%	2	7%	1.00
Processing speed	1	8%	0	0%	0.293
Total	1	8%	2	7%	1.00

Note. Dissociation was defined as a case in which two requirements were met: (1) being outside a 95% density ellipse of the scatterplot between receptive and expressive functioning scores and (2) having a difference of more than 1 standard deviation in the two scores. Receptive and expressive functioning were assessed with British Picture Vocabulary and Picture Naming for vocabulary, the Test for Reception of Grammar and the Grammatic Closure subtest from the ITPA for grammar, the mean of four Phonological awareness tests and the Children’s Test of Non-Word Repetition for phonological skills, and the Child Language and Learning’s speed of processing tests and the Rapid Automized Naming task for processing speed. The total is the combination of the previous four domains. Data are based on 50 multiple imputed datasets for 0.7% missing data. *p*-values are based on Fisher’s exact chi-square test.

## Data Availability

The data are available in the services for sensitive data at the University of Oslo and can be obtained by contacting the first author.

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
