# Peer review of "The Association between Difficulties with Speech Fluency and Language Skills in a National Age Cohort of Children with Down Syndrome"

_brainsci, 2021, doi:10.3390/brainsci11060704_

Round 1

Reviewer 1 Report

This is a well written and competent paper addressing an important issue - the effect of speech fluency difficulties on language development for children with Down syndrome. It highlights the high incidence of speech fluency issues for this group compared to NVMA matched TD children. One strength is the size of the cohort within a narrow age range as so many studies end up with a wide age range of participants with Down syndrome, making interpretation difficult. The authors acknowledge the limitations of parent report of speech fluency issues but it can be argued that parents do see the daily effects of the issues and their estimates may turn out to be highly correlated with objective assessments carried out by SLT. This does need to be confirmed in future studies.  I would like to see information on how the accuracy of production was dealt with for all the expressive language and expressive phonology assessments added.  This is important in the light of no apparent effect of speech fluency issues when the receptive expressive gap increased. The demonstration of the association of degree of speech fluency  difficulties with poorer language skills, specifically vocabulary and phonological skills, has important practical implications even though effective interventions have not yet been demonstrated. The next step is to look at early speech perception and speech production development in TD infants as much happens in early months, in order to develop and evaluate interventions. 

Reviewer 2 Report

In this study, the authors compared speech fluency of 6-year-old children with Down syndrome to that of typically developing 3-year-old children who were matched on nonverbal MA (using the Block Design subtest of the WPPSI-III). Speech fluency was assessed via parental report. Subsequently, they evaluated the association between speech fluency difficulties and several specific language skills of the participants with Down syndrome (vocabulary, grammar, phonology, and speed of processing). Results indicated a very high level of fluency problems in the children with Down syndrome (over 70%) which was significantly greater than the typically developing children (15%), with the children with Down syndrome exhibiting more severe difficulties. There was an association between fluency and language skills such that vocabulary and phonological skills, in particular, were significantly associated with fluency when nonverbal ability was controlled. No association was observed between fluency and the dissociation with expressive vs receptive language.

Overall, the MS is very well written. Taken together, the results are relatively clean and provide some important data of interest to researchers and practitioners who work with children with Down syndrome. I have one primary concern with the general method and presentation that I would lie to see addressed by the authors. Specifically, my concern is with the sample and how well the results are likely to generalize to a broader group of children with Down syndrome. The sample used in this study were recruited from children about to enter school for the first time. Hence, all were approximately the same age and all were without any degree of formal schooling. We do not know if any or all of the participants with Down syndrome had any language training or therapy. I suppose it is possible that some or all may have had some language therapy, but we do not know. In any event, this is a one-time sample with highly unique characteristics.

What I would like to see is a two-fold approach to addressing these unique sample characteristics. First, highlight the importance of this knowledge with this particular sample of children who are about to enter formal schooling for the first time. I think this should be identified in the introduction as well as the discussion. In essence, it is important to know that children with Down syndrome entering school exhibit this pattern of language skills. Second, speculate about how this pattern may or may not change as a function of formal language instruction. Data available for older children and adolescents may provide some direction for this. At the very least, there needs to be some discussion about how the choice of sample may have impacted the results and may be limited to children who have had these same, relatively limited language opportunities.

I was also concerned that fluency was based on parental report as I read the MS– but I believe that the authors identified and addressed that issue as well as possible. But, it may be worthwhile to point out more directly that some of the observed associations with fluency may be due to parents responding more about language skills more generally than would experts.
